# Validating Accuracy of an Internet-Based Application against USDA Computerized Nutrition Data System for Research on Essential Nutrients among Social-Ethnic Diets for the E-Health Era

**DOI:** 10.3390/nu14153168

**Published:** 2022-07-31

**Authors:** Ya-Ling Yang, Hsiao-Ling Yang, Joyce D. Kusuma, Shyang-Yun Pamela Koong Shiao

**Affiliations:** 1School of Nursing, College of Medicine, National Taiwan University, Taipei 10051, Taiwan; ylyang@ntu.edu.tw (Y.-L.Y.); slyang@ntu.edu.tw (H.-L.Y.); 2Heritage Victor Valley Medical Group, Big Bear Lake, CA 92315, USA; 3Mt San Antonio Gardens, Pomona, CA 91767, USA; 4Center for Biotechnology and Genomic Medicine, Medical College of Georgia, Augusta University, Augusta, GA 30912, USA

**Keywords:** personalized nutrition, internet-based applications, mobile app, Nutrition Data System for Research (NDSR), social-ethnic diets, e-Health, generalized regression, dietary record, agreement and bias

## Abstract

Internet-based applications (apps) are rapidly developing in the e-Health era to assess the dietary intake of essential macro-and micro-nutrients for precision nutrition. We, therefore, validated the accuracy of an internet-based app against the Nutrition Data System for Research (NDSR), assessing these essential nutrients among various social-ethnic diet types. The agreement between the two measures using intraclass correlation coefficients was good (0.85) for total calories, but moderate for caloric ranges outside of <1000 (0.75) and >2000 (0.57); and good (>0.75) for most macro- (average: 0.85) and micro-nutrients (average: 0.83) except cobalamin (0.73) and calcium (0.51). The app underestimated nutrients that are associated with protein and fat (protein: −5.82%, fat: −12.78%, vitamin B12: −13.59%, methionine: −8.76%, zinc: −12.49%), while overestimated nutrients that are associated with carbohydrate (fiber: 6.7%, B9: 9.06%). Using artificial intelligence analytics, we confirmed the factors that could contribute to the differences between the two measures for various essential nutrients, and they included caloric ranges; the differences between the two measures for carbohydrates, protein, and fat; and diet types. For total calories, as an example, the source factors that contributed to the differences between the two measures included caloric range (<1000 versus others), fat, and protein; for cobalamin: protein, American, and Japanese diets; and for folate: caloric range (<1000 versus others), carbohydrate, and Italian diet. In the e-Health era, the internet-based app has the capacity to enhance precision nutrition. By identifying and integrating the effects of potential contributing factors in the algorithm of output readings, the accuracy of new app measures could be improved.

## 1. Introduction

For personalized precision nutrition, adequate intakes of macro-and micro-nutrients could reduce disease risks of obesity, metabolic disease, cardiovascular disease, and cancer across an individual’s the life span [1,2,3,4]. Hence, accurate measurements of essential micro-nutrients, in addition to macro-nutrients, are vital in achieving precision nutrition [5,6]. Methyl donors including micro-nutrients such as vitamins (folate B9, cobalamin B12) and amino acids are crucial for DNA methylation in nutrigenomics pathways for personalized precision nutrition [7,8,9,10,11]. Various internet-based applications (apps) are rapidly developing in the e-Health era [12,13,14], that are accessible and convenient for assessing nutrient intake in real-time [15,16,17,18]. However, compared to established dietary measures, the accuracy of apps is yet to be examined for essential micro-nutrients in addition to macro-nutrients, with records of 1–3 day diaries and longer durations including a food frequency questionnaire (FFQ) [12,13,19].

The validity of four apps was examined against the established 1–3 day diaries of the Nutrition Data System for Research (NDSR), and the agreement using intraclass correlation coefficient (ICC) for macro-nutrients varied, ranging from excellent (0.9–1.0) for calories, and moderate to excellent for other nutrients (carbohydrate: 0.65–0.99; protein: 0.64–1.00; fat: 0.52–1.0; fiber: 0.52–0.99; cholesterol: 0.77–1.0; sodium: 0.69–0.9) [13,19]. Using Pearson’s correlation, five apps validated against NDSR presented similar findings ranging from excellent (0.89–0.94) for calories, and moderate to excellent for other nutrients (carbohydrate: 0.73–0.94; protein: 0.83–0.96; fat: 0.75–0.88; fiber: 0.74–0.92; cholesterol: 0.62–0.81; sodium: 0.60–77) [19]. The differences between the apps and NDSR ranged from −7.92 to 8.35 for calories [13], with an underestimation of 7–41% for protein, fat, fiber, cholesterol, and sodium by the apps [19]. However, prior validation studies did not include essential micro-nutrients, which are critical in nutrigenomics pathways for personalized nutrition across various populations consuming various diet types [13,20]. Thus, macro- and micro-nutrients need to be examined across various social-ethnic diets for the apps against the established method.

Cultural ethnic inheritance, social change, and individual preferences influence human dietary habits [21], where traditional ethnic diets include examples of convenient American, Mediterranean, Italian, Chinese, and Japanese diets [13,19,20,21,22,23,24]. Additionally, fast foods [25,26], high school foods [26], smoothies [27,28,29], pure liquids [27,30], and dozens of other diet types emerged following urbanization and growing aging populations. While Mediterranean and Japanese diets are beneficial to prevent cardiovascular disease; some Western diets are loaded with empty calories leading to metabolic disorders [31,32], and rice-based ethnic diets contain insufficient zinc and iron to meet the daily requirements [33]. While smoothies have been commonly loaded with vegetables and fruits, they might increase daily caloric intake via excess sugar and fat intake if unchecked, with insufficient essential nutrients [33,34]. Hence, we included pure liquids and smoothies [27,33,34,35] as they are common additions to social-ethnic human diets in various healthcare settings. Additionally, convenient diets [25,26] and various social-ethnic diets are common in a social-ethnic context, thus they were also included in this study [21,36,37].

The Bland-Altman analysis is a commonly used method to examine the validity of a new measure against another measure [13,19,38,39,40,41]. Additionally, machine-learning-based, artificial-intelligence (AI), generalized regression (GR) models could be employed to identify the source factors that could contribute to the differences between measures [7,42], with enhanced prediction accuracy [7,12]. A previous study noted that the source factors that could contribute to the differences between the two measures of an app and FFQ included caloric ranges; the difference between two measures for carbohydrates, protein, fat; and diet types [12]. Therefore, in this study, we further validated the app against NDSR, to identify factors that could contribute to the differences between two measures for essential nutrients including various diet types, integrating AI method to confirm the source factors to the differences between the two measures.

## 2. Materials and Methods

We evaluated the essential nutrients from 131 social-ethnic diets taken by various vulnerable populations, continued from our series of studies [7,12,42]. Under-reporting is a common human behavior in human studies in recording dietary intake [13,43,44]; thus, validation of dietary intake with model social-ethnic diets as recipes or menus can offer precise and controlled nutrient intake for individuals to follow and consume [12,45]. We examined four groups of diets, including: (1) pure liquids [27,30,46]; (2) convenient diets [25,26,34]; (3) ethnic diets of Western and Eastern origins [21,24]; and (4) smoothies-added [27,28,29]. We selected various diets to enhance the delivery of nutrients based on possible forms of liquids and solid foods. To add variations to the baseline daily recipes of these diets, excess calories, proteins, vegetables, fruits, and fats were included.

We evaluated essential nutrients of these diets based on the United States National Institute of Health’s dietary reference intake [47] using both NDSR and the app based on 3-day 24-h dietary diaries throughout the week. Macro-nutrients included total calories, carbohydrates, protein, total fat, saturated fat, cholesterol, and fiber [12]. Micro-nutrients included vitamins: thiamin B1, riboflavin B2, niacin B3, pyridoxine B6, folate B9, and cobalamin B12, A, C, D, E; amino acids: methionine, glycine [47]; choline; and minerals: zinc, calcium, magnesium, iron, and sodium. Foods that are rich in carbohydrates also commonly contained fiber, vitamins A, B9, and C. Whereas, foods that are meat-based are rich in protein, fat, saturated fat, cholesterol, vitamin B12, methionine, glycine, and zinc [47].

### 2.1. Social-Ethnic Diets

Details of these diets were included in a prior report [12] are summarized in the following sections. Liquids are necessary for human hydration and gut motility [27,28]. Pure liquid diets have been commonly recommended for frail elderly individuals and patients with terminal illness and gastrointestinal dysfunction following surgery [27,30]. Examples of liquid diets comprise: various liquids loaded with minerals, fruits, vegetables, and soups. Convenient diets in modern industrial societies include canned-foods, café diets, and fast foods [25,26]. The canned-foods are comprised of items such as soups, various vegetables, noodles, and fish. The café diets include foods offered at high schools, frequently were loaded with sodium and saturated fat [34] such as chicken wings, chicken tenders, grilled cheese sandwiches, mini cheeseburgers, pizza, cheese sticks, hot dogs, and corn dogs. The fast-foods also contain high fat, saturated fat, sodium, sugar, and empty calories [25,26,48], such as fried meats, fries, biscuits, sandwiches, wraps, and hash browns [12,25,26].

Various ethnic diets include Western influences from American, Mexican, Italian, and Mediterranean diets, and Eastern influences from Japanese, Chinese, and Korean diets [21,24,37]. The American diet embodies fried foods and salads [31]. The Mexican diet contains grain- and bean-based foods with meats and chili soups. The Italian diet includes grains and vegetables. The Mediterranean diet contains plenty of fruits, vegetables, whole grains, nuts, and olive oil, with some meat [37]. The Japanese diet includes seafoods, soups, and tempura [49]. The Chinese diet embodies meats, tofu, eggs, noodles, and rice [50]. The Korean diet includes rice, meats, mixed bowls of vegetables, and soups [21,24]. A smoothie-added diet could be used when people perform physically strenuous activities that require additional hydrations or nutrients. The smoothie-added diet includes a variety of fruits and vegetables added to the base diet, with ethnic-based ingredients and multi-grains [27,28,29,51].

### 2.2. Dietary Measures and Nutrient Intake

We analyzed 3-day dietary intake of nutrients from various social-ethnic diets using both NDSR and an app. We used NDSR software version 2015 based on published values in the USDA nutrient database (NDSR, Minnesota University, MN, USA; http://www.ncc.umn.edu/products/nutrients-nutrient-ratios-and-other-food-components/primary-energy-sources/ (assessed on 1 June 2022)), which was initiated in 1974 by the National Heart, Lung, and Blood Institute (NHLBI) [7]. NDSR was developed from 19,000 foods that embody an array of ethnic foods and common menu items for analysis of 178 nutrients [12,52,53,54]. NDSR is widely used for nutrient analysis based on the 24-h diary, recipes, and menus, across populations in many countries with different diet types [12,52,53,54]. NDSR provided detailed reports on the quantity of macro- and micro-nutrients [55]. The app was developed by a nutrigenomics company that specialized in the internet-based mobile app to assess daily dietary intake (GB HealthWatch, San Diego, CA, USA, https://healthwatch360.gbhealthwatch.com, accessed on 1 June 2021) [12,23,56]. The app, based on daily food logs, could extricate 30 essential nutrients. Data accuracy was confirmed by team members before analysis.

### 2.3. Data Analysis

We analyzed data with JMP version 15.0.0 statistical software [57,58,59,60] (SAS Institute Inc., Cary, NC). We evaluated the agreement and bias between the apps and NDSR. Means and standard deviations (SD) for nutrients [61] were calculated for both measures. The agreement between the two measures for the parameters were analyzed using ICCs and mean % differences, and bias with standard errors (SE). Pairwise correlations for ICCs (*r*) between the two measures presented the strengths of association (excellent: ≥0.9, good: 0.75 ≤ 0.90, moderate: 0.50 ≤ 0.75, poor: <0.50) [39,55,62]. The Bland–Altman plots presented differences with limits of agreement (LoAs: mean differences ± 2 SD) between the two measures [12,63,64], with good agreement at 95% or greater [65,66,67]. The alpha for the significance level for all analyses was set at 0.05.

The analytics were described in detail in earlier studies [7,12,42], and summarized in the following. We employed GR models with AI-based machine learning methods to identify the source of differences between the two measures by progressively incorporating related factors in the analysis [7,12,42]. The source factors that could contribute to the differences between the two measures included (1) caloric ranges of total calories (<1000, 1000–2000, or >2000); (2) effects of differences from carbohydrates, protein, fat; and (3) diet types [12]. JMP software provided logistic regression (LR) as the default model to explain the baseline dependent variables. Following LR, we selected the Elastic Net estimated method (Leave-One-Out (LOO) and Validation Column for confirmatory analysis to predict the accuracy with smaller misclassification rate for minimal prediction error by avoiding over-fitting [7,12,42,68]. Elastic Net models were employed for effective use in handling complex multiple domains in the datasets, balancing potential interactions [69]. Both LOO and AICc validation are effective for small sample sizes and handling multiple domains [7,59]. LOO is used to select source factors within domains of caloric ranges, macro-nutrients, and diet types [70,71]. We used AICc validation columns to confirm how well the model fits with unbiased best model prediction [70], with 80/20 randomized split for training and validation set for predictive modeling. The selection of the best model is based on AICc (lower score is fitter, more precise), misclassification (smaller is more accurate), and area under the Receiver Operating Characteristics (ROC) curve (AUC, higher coverage is better) [42,60]. The interaction profilers were used to visualize the significant interactions, if found could be included in the final model [70,71].

## 3. Results

### 3.1. Agreement and Difference: The Bland-Altman Method

We validated 131 diets using the Bland-Altman method to assess the agreement (ICCs, % differences) and bias (SE) between the app and NDSR per three caloric ranges (<1000, 1000–2000, and >2000) and selected essential nutrients (Table 1) using 5–10% differences as agreement criteria [62]. The ICCs between the two measures for total calories were good for all diets together (*r* = 0.85), but moderate and worst for outliers of >2000 caloric range (<1000 calories: 0.75, 1000–2000 calories: 0.76, >2000 calories: 0.57 (95% Confidence Interval: 0.14–0.82)). The ICCs between the two measures were good for most other nutrients (average for macro- 0.85 and 0.83 for micro-nutrients), but moderate for cobalamin (0.73) and calcium (0.51) (all *p* < 0.001).

Compared to NDSR, the app presented acceptable differences (<5%) on total calories with the caloric range of <1000 and 1000–2000, but underestimated >2000 calories (>10%). For macro-nutrients, the app underestimated total calories and protein (>5%), fat and saturated fat (>10%); overestimated fiber (>5%); but presented acceptable differences with carbohydrate and cholesterol (<5%). For micro-nutrients, the app underestimated methionine (>5%); vitamin B12, glycine, zinc, and sodium (>10%); overestimated vitamin B1 and B9 (>5%), as well as vitamin A (>10%); but presented acceptable differences with vitamin B2, B3, B6, C, D, E, choline, calcium, magnesium, and iron (<5%). Thus, we found good agreement between the two measures on some micro-nutrients but not protein-based micro-nutrients that are essential in personalized precision nutrition.

The bias (SE) increased with increased caloric ranges (1.26 for <1000 calories, 1.65 for 1000–2000 calories, and 6.06 for >2000 calories). The bias was greater (>2) for vitamins A, C, E, and calcium when compared to all other nutrients including fiber and sodium (ranges 1.14–1.83). Furthermore, we used Bland–Altman plots to visualize the differences and LoAs between the app and NDSR. We presented side-by-side comparison of correlations (left panel) and Bland–Altman plots (right panel) on exemplary macro-nutrients of calories (Figure 1a,b), fat (Figure 2a,b), protein (Appendix A), carbohydrate (Appendix A); and micro-nutrients of folate (Figure 3a,b) and cobalamin (Figure 4a,b). Correlation plots presented good agreement between the two measures on all nutrients as presented in these figures. Bland–Altman plots displayed a wide variability of differences for both macro- (calories, fat, protein, carbohydrate) and micro-nutrients (folate, cobalamin). Furthermore, Bland–Altman plots illustrated the underestimation by the app compared to NDSR on total calories (−5.2%, Figure 1b), fat (−12.78%, Figure 2b), protein (−5.82%, Appendix A), and cobalamin (−13.59%, Figure 4b); but overestimation on carbohydrate (0.93%, Appendix A) and folate intake (9.06%, Figure 3b).

Additionally, we assessed the differences between the app and NDSR for various nutrients in association with caloric ranges and diets (Table 2 and Appendix A). To continue findings in Table 1 for the underestimation of total calories with >2000 calories, the app underestimated total calories with canned food, Mediterranean, Chinese, and smoothie-added diets (>5%), and fast food (>10%); but presented acceptable differences for other diets (<5%). For carbohydrates, the app presented acceptable differences with <1000 calories (<5%), but overestimated with 1000–2000 calories (>5%) and underestimated with >2000 calories (>5%). Additionally, for carbohydrates, the app underestimated with smoothie-added diets (>5%); overestimated with canned food and high school (>5%), and Mexican diets (>10%); but presented acceptable differences (<5%) with other diets (pure liquid, fast food, Italian, Mediterranean, American, Japanese, Chinese, and Korean). For protein, the app underestimated with <1000 calories (>5%) and >2000 calories (>10%), but presented acceptable differences with 1000–2000 calories (<5%); and underestimated with diets including pure liquids, Mexican, and smoothie-added diets (>5%), Mediterranean, Chinese, and Korean diets (>10%); but presented with acceptable differences with other diets of canned-food, high school, fast-food, Italian, American, and Japanese (<5%). The app underestimated fat with all caloric ranges (>10%); with diets, including high school, Mexican, Italian, Mediterranean, and Japanese diets (>5%); as well as canned food, fast food, Chinese, Korean, and smoothie-added diets (>10%); but presented acceptable differences with pure liquids and American diets (<5%). For folate, the app presented overestimation with 1000–2000 calories (>5%), <1000 calories (>10%), and underestimation with >2000 calories (>5%); overestimation with Italian, Mediterranean, American, Japanese, and Korean diets (>5%); pure liquid, canned food, high school, fast food, and Mexican diets (>10%); but acceptable differences with Chinese and smoothie-added diets (<5%). For cobalamin, the app underestimated with all caloric ranges (>10%); underestimated canned food and Italian (>5%) diets; pure liquid, high school, Mexican, Mediterranean, Chinese, Korean, and smoothie-added diets (>10%); overestimation with Japanese (>5%) diet; but presented acceptable differences with American and fast-food diets (<5%). Additionally, the app underestimated nutrients that are associated with protein- and fat-based intake, including saturated fat, cholesterol, methionine, choline, glycine, vitamins B12 and E; but overestimated nutrients that are associated with carbohydrate-based intake, including fiber, and vitamins B9, A, and C (Appendix A).

### 3.2. Predictive Modeling for the Difference between the Internet-Based App against NDSR: Generalized Regression Analysis

For predictive modeling, we progressively examined significant factors per domains of caloric ranges (coded as one of the three versus the other two categories for <1000, 1000–2000, and >2000), energy-producing macro-nutrients, and diets. We included the significant factors of all domain factors in the final combined model (Appendix A progression examples for total calories, Appendix A for folate, and Appendix A for cobalamin). For total calories, differences of <1000 over other caloric ranges, fat, and protein were significant contributing factors to the difference between the app and NDSR (misclassification 0.19, AICc 30.29, and AUC 0.89) (Table 3, baseline LR model on the left panel and GR model validation on the right panel). As an example, Figure 5 illustrates the AUC curve for total calories with 89% for both sensitivity and specificity for the accuracy of the selected model [71]. Through the progressive analysis, we observed a higher AICc and less precise model by including an additional factor, the Chinese diet, thus producing a less favorable model than the selected model (Appendix A).

We selected folate and cobalamin as the most representative essential micro-nutrients in precision nutrition for more detailed presentation. Factors that contributed to the differences in folate between the two measures included caloric range (<1000 versus the other two categories), carbohydrate, and Italian diet (misclassification 0.30, AICc 38.5, and AUC 0.90) (Table 4). Through the progression analysis, we noted a higher AICc and AUC by including an additional factor, the Chinese diet, thus, this factor was not included (Appendix A). For cobalamin, significant factors that contributed to the differences between the two measures included protein, and American and Japanese diets (misclassification 0.26, AICc 37.7, and AUC 0.81) (Table 5). In the progression analysis, we observed a higher AICc by including additional factors of Chinese and Mexican diets, which presented less favorable models than the selected model; thus, they were not included in the final model (Appendix A).

We also assessed significant factors between the app and NDSR for other nutrients in the progressive analysis, and summarized the final models in the Appendix A (carbohydrate, protein, fat, saturated fat, cholesterol, and fiber in Appendix A; thiamin, riboflavin, niacin, pyridoxine, choline, glycine, and zinc in Appendix A; and vitamins A, C, D, and E, and calcium, magnesium, iron, and sodium in Appendix A). Significant factors that contributed to the differences between the two measures for carbohydrate included calories, fiber, and Italian diet; for protein: caloric range (1000–2000 versus others), total calories, cholesterol, and canned foods; for fat: saturated fat, cholesterol, and fast food; for saturated fat: total calories, fat, and Korean diets; for cholesterol: fat and pure liquids; for fiber: caloric range of 1000–2000, carbohydrate, and canned-foods (misclassification 0.11–0.22, AICc 24.0–33.7, AUC 0.82–0.98). Similarly, significant factors for thiamin included total calories, fiber, high school, and Chinese diet; for riboflavin: carbohydrate, protein, and high school diets; for niacin: total calories and Chinese diet; for pyridoxine: total calories and canned-food; for choline: total calories, and canned-food; for glycine: protein, canned-food, and Japanese diet; and for zinc: total calories, protein, canned-food, and Japanese diet (misclassification 0.04–0.22, AICc 19.8–35.7, AUC 0.88–0.97). Additionally, significant factors for vitamin A included protein, and Korean and smoothie-added diets; for vitamin D: cholesterol, canned-food, and Mediterranean diet; for vitamin E: fat, fast food, and Italian diet; vitamin C: fiber, fast food, and Mexican diet; for calcium: cholesterol, and Mexican and Chinese diets; for magnesium: total calories, protein, fiber, canned-food, and high school food; for iron: protein, high school food and Mediterranean diet; for sodium: saturated fat, high school food, and Mexican diet (misclassification 0.11–0.33, AICc 31.2–41.7, AUC 0.73–0.90). The canned-food diet type was a common contributing factor to the differences between the two measures for protein, fiber, pyridoxine, choline, glycine, zinc, vitamin D, and magnesium (Appendix A). We did not explicitly test the model for methionine, as methionine is purely dependent on protein for calculating the intake level. The interaction profiler plots did not present any significant three-way interactions for inclusion in the final models for all nutrients.

## 4. Discussion

For precision nutrition, we validated the accuracy of an internet-based app in assessing macro- and micro-nutrients in various social-ethnic diets, against 3-day NDSR in this study, in addition to previous validation against FFQ [13,14,19]. Compared to previous findings with excellent ICCs on calories (0.9–1) between the app and NDSR [13,19], we found good agreement (≥0.75) between the two measures for total calories (0.85). We further identified good agreement (≥0.75) between the two measures for total calories with caloric ranges of 1000–2000 (0.76), but moderate agreement with caloric ranges of <1000 (0.75) and >2000 (0.57). Compared to previous findings of moderate to excellent agreement for other major nutrients [13,19], the ICCs in this study between the two measures for most macro-nutrients were good (≥0.75) (average: 0.85 for all macro-nutrients). We further demonstrated the good agreement between the two measures using ICCs for most micro-nutrients (average 0.83), but found only moderate agreement for cobalamin (0.73) and calcium (0.51) (all *p* < 0.001).

In comparison to NDSR, the app underestimated protein and fat-based nutrients (protein: −5.82%, fat: −12.78%, vitamin B12: −13.59%, methionine: −8.76%, zinc: −12.49%), while it overestimated carbohydrate-based nutrients (fiber: 6.7%, B9: 9.06%). Thus, we found similar underestimation of protein and fat for macro-nutrients between the app and NDSR, as in a previous study [19]. Contrary to the validation against FFQ with the acceptable estimation of vitamins B1 (2.46%) and B9 (3.24%), this study demonstrated that the app overestimated vitamins B1 (6.48%) and B9 (9.06%) against NDSR. For choline, the app presented acceptable estimation (−4.51%) using 5% criteria, against NDSR, while it was underestimated with the app against FFQ (−6.23%). The bias (SE) was small for all three caloric ranges (0.63–0.88), between the two measures of the app and NDSR; which are smaller and more precise in this study compared to the findings in a prior study with the app against FFQ (1.44–5.91) [12,19]. The smaller bias could be due the same recording duration of 3-day diaries with both the measures in this study, as contrary to a longer duration in measure with FFQ for the prior study. The correlations between the two measures in this study were strong ≥0.70 for most nutrients (average for macro- 0.85 and 0.82 for micro-nutrients) except for calcium (0.51), that are of similar findings as in a prior study with app being validated against FFQ [12].

We identified the factors that contributed to the differences between the two measures. For total calories, the sources that contributed to the difference between the two measures included caloric range (<1000 versus others), fat, and protein. For folate, the sources included caloric range (<1000 versus others), carbohydrate, and Italian diet. Additionally, for cobalamin, the sources included protein, and American and Japanese diets. Therefore, continuing from a previous study [12], caloric ranges and various diets could be used to identify the sources that might contribute to the differences between the two measures, when validating a new measure such as an internet-based app against established dietary measures.

In summary, fat and protein continued to be the major nutrient sources of differences in the validation of the app against established dietary measures, when compared to NDSR as well as FFQ [12]. The confirmatory predictive modeling further substantiated this result, with specific caloric ranges and diet types also as sources of differences. Hence, these source factors could be used to adjust the algorithm of readings in the development of the internet-based app. With advanced technology and AI analytics in the e-Health era, these apps have the capacity to enhance precision nutrition, by integrating potential contributing factors in the development of new accurate measures. Various diets across populations and related factors should be included in future studies to further validate dietary measures.

## Figures and Tables

**Figure 1 nutrients-14-03168-f001:**
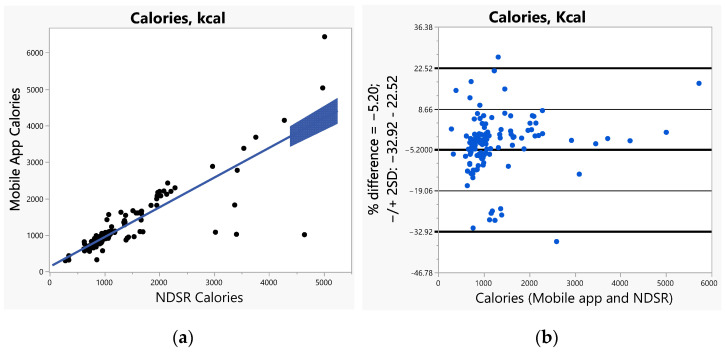
(**a**) Correlation, (**b**) Bland-Altman plots between the internet-based application and Nutrition Data System for Research (NDSR) on total calories.

**Figure 2 nutrients-14-03168-f002:**
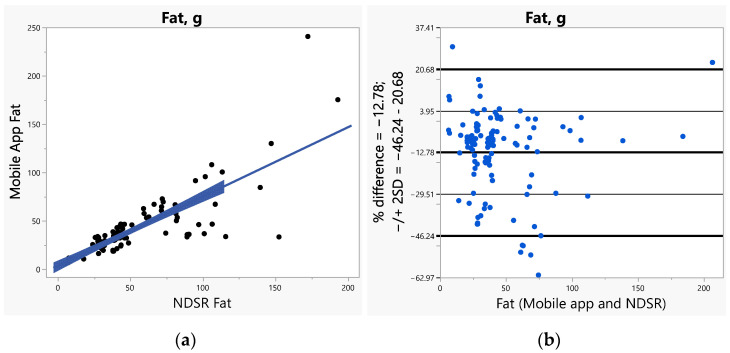
(**a**) Correlation, (**b**) Bland-Altman plots between the internet-based application and Nutrition Data System for Research (NDSR) on fat.

**Figure 3 nutrients-14-03168-f003:**
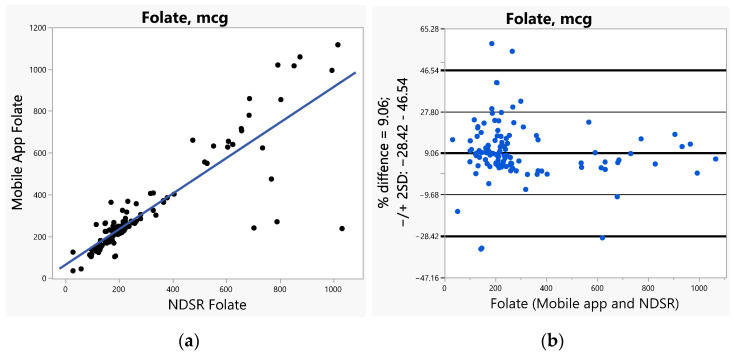
(**a**) Correlation, (**b**) Bland-Altman plots between the internet-based application and Nutrition Data System for Research (NDSR) on folate.

**Figure 4 nutrients-14-03168-f004:**
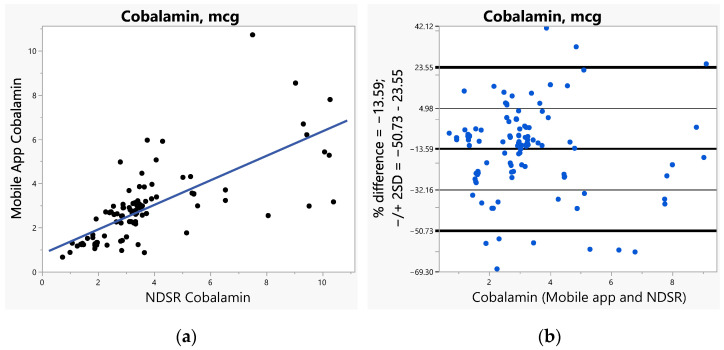
(**a**) Correlation, (**b**) Bland-Altman plots between the internet-based application and Nutrition Data System for Research (NDSR) on cobalamin.

**Figure 5 nutrients-14-03168-f005:**
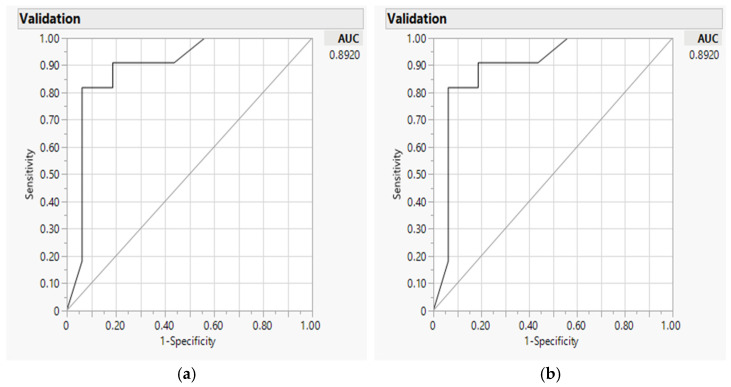
Predicting accuracy of total calorie analyses using internet-based application against NDSR: Area under the receiver operating characteristic curve (AUC) for baseline logistic regression model (**a**) and Elastic Net with validation model (**b**).

**Table 1 nutrients-14-03168-t001:** Agreement and bias for the internet-based application (App) against Nutrition Data System for Research (*n* = 131).

Parameters	*r* (95% CI) **	% Difference M ± SD	NDSR M ± SD	App M ± SD	SE	±2 SD%
Calories (kcal)	0.85 (0.80, 0.89)	−5.20 ** ± 13.86	1333 ± 891.8	1215 ± 853.6	1.21	95.42
<1000 (*n* = 63)	0.75 (0.62, 0.84)	−4.26 ** ± 10.01	806.5 ± 147.1	756.6 ± 162.5	1.26	95.65
1000–2000 (*n* = 50)	0.76 (0.60, 0.85)	−4.10 * ± 11.67	1330 ± 302.9	1255 ± 346.3	1.65	92.31
>2000 (*n* = 18)	0.57 (0.14, 0.82)	−11.56 * ± 25.71	3183 ± 1042	2708 ± 1410	6.06	95.00
Carbohydrate (g)	0.85 (0.80, 0.89)	0.93 ± 16.22	180.60 ± 141.0	178.4 ± 136.1	1.42	92.37
Protein (g)	0.85 (0.80, 0.89)	−5.82 ** ± 13.05	52.20 ± 33.25	47.35 ± 33.10	1.14	93.89
Fat (g)	0.81 (0.75, 0.86)	−12.78 ** ± 16.73	47.88 ± 33.56	37.76 ± 29.57	1.46	93.13
Sat Fat (g)	0.84 (0.79, 0.89)	−13.77 ** ± 16.70	14.26 ± 9.81	11.54 ± 9.35	1.46	94.66
Cholesterol (mg)	0.88 (0.84, 0.91)	−4.86 ** ± 15.57	184.2 ± 117.3	175.4 ± 121.7	1.36	93.89
Fiber (g)	0.85 (0.80, 0.89)	6.70 ** ± 20.93	17.99 ± 19.30	18.46 ± 18.91	1.83	92.37
Thiamin (mg)	0.85 (0.80, 0.89)	6.48 ** ± 14.00	1.04 ± 0.71	1.12 ± 0.73	1.22	94.66
Riboflavin (mg)	0.86 (0.81, 0.90)	0.19 ± 13.72	1.26 ± 0.85	1.23 ± 0.81	1.20	93.89
Niacin (mg)	0.86 (0.80, 0.90)	0.51 ± 14.15	14.43 ± 9.52	14.34 ± 10.40	1.24	93.13
Pyridoxine (mg)	0.86 (0.80, 0.90)	−4.38 ** ± 15.17	1.74 ± 1.46	1.60 ± 1.42	1.33	94.66
Folate (mcg)	0.87 (0.81, 0.90)	9.06 ** ± 18.74	280.2 ± 219.6	302.3 ± 215.9	1.64	92.37
Cobalamin (mcg)	0.73 (0.64, 0.80)	−13.59 ** ± 18.57	3.42 ± 2.05	2.70 ± 1.56	1.62	92.37
Methionine (g)	0.84 (0.79, 0.89)	−8.76 ** ± 13.20	1.17 ± 0.72	1.03 ± 0.74	1.15	93.89
Choline (mg)	0.82 (0.76, 0.87)	−4.51 ** ± 18.55	263.4 ± 177.0	240.1 ± 161.9	1.62	95.42
Glycine (g)	0.83 (0.76, 0.87)	−10.47 ** ± 14.80	2.26 ± 1.42	1.93 ± 1.45	1.29	92.37
Vitamin A (IU)	0.86 (0.81, 0.90)	28.00 ** ± 31.32	13,294 ± 17,265	16,373 ± 16,564	2.74	96.95
Vitamin C (mcg)	0.88 (0.83, 0.91)	2.41 ± 24.76	149.4 ± 158.3	149.1 ± 167.3	2.16	93.13
Vitamin D (mcg)	0.89 (0.84, 0.92)	0.72 ± 14.53	3.90 ± 2.14	3.91 ± 2.20	1.27	95.42
Vitamin E (mcg)	0.82 (0.75, 0.87)	−0.10 ± 24.31	6.68 ± 5.34	6.13 ± 4.91	2.12	92.37
Zinc (mg)	0.83 (0.77, 0.88)	−12.49 ** ± 15.82	7.64 ± 4.55	6.19 ± 3.92	1.38	92.37
Calcium (mg)	0.51 (0.37, 0.62)	1.46 ± 31.82	548.4 ± 335.5	570.8 ± 389.0	2.78	93.13
Magnesium (mg)	0.86 (0.81, 0.90)	1.40 ± 15.69	211.6 ± 162.7	210.3 ± 159.6	1.37	93.89
Iron (mg)	0.86 (0.81, 0.90)	3.11 ± 18.52	8.98 ± 6.02	8.99 ± 5.66	1.62	94.66
Sodium (mg)	0.75 (0.67, 0.82)	−19.78 ** ± 18.84	2867 ± 1869	2002 ± 1266	1.65	94.66

Note: NDSR: Nutrition Data System for Research; *r*: pairwise for intraclass correlation coefficient; CI: confidence interval; M: mean; SD: standard deviation; * *p* < 0.05; ** *p* < 0.001.

**Table 2 nutrients-14-03168-t002:** Differences between the internet-based application and Nutrition Data System for Research per domains of caloric ranges and various diets for essential macro-and micro-nutrients (*n* = 131).

Parameters (*n*)	Calories, kcal %Diff M ± SD	Carb, g %Diff M ± SD	Protein, g %Diff M ± SD	Fat, g %Diff M ± SD	Folate, mcg %Diff M ± SD	Cobalamin, mcg %Diff M ± SD
**Caloric range**						
<1000 (63)	−4.26 **± 10.01	−0.70 ** ± 12.55	−6.27 ** ± 11.35	−10.58 **± 13.73	12.79 **± 18.86	−13.43 **± 18.61
1000–2000 (50)	−4.10 **± 11.67	5.15 **± 11.60	−3.03 **± 9.04	−14.34 **± 17.19	9.54 **± 10.94	−12.27 **± 15.28
>2000 (18)	−11.56 ± 25.71	−9.98 ± 29.51	−12.03 * ± 23.09	−16.16 **± 23.74	−5.35 ± 27.79	−17.83 * ± 26.04
**Diet types**						
Pure Liquid (8)	−0.38 ± 7.32	2.88 ± 4.52	−9.53 ± 17.28	−0.66 ± 17.65	20.05 ± 38.13	−22.04 ** ± 23.39
Convenient Diet (30)	−8.30 ** ± 12.81	4.18 ** ± 9.94	1.85 ± 10.41	−23.65 ** ± 20.16	20.47 ** ± 15.55	−8.11 ** ± 12.91
Canned Food (10)	−6.83 * ± 9.16	5.60 ** ± 4.15	3.84 ** ± 4.35	−27.27 ** ± 17.56	30.51 ** ± 15.86	−8.96 ** ± 1.84
High School (10)	0.55 ± 3.84	5.50 ** ± 2.64	0.42 ± 8.33	−5.32 * ± 5.35	19.10 ** ± 6.64	−12.51 * ± 15.18
Fast Food (10)	−18.61 ** ± 14.93	1.43 ± 16.78	1.30 ± 15.94	−38.36 ** ± 18.54	11.81 ** ± 16.93	−2.87 ± 15.82
Ethnic Food (71)	−3.22 ** ± 9.90	2.01 ± 13.96	−7.40 ** ± 8.35	−9.14 ** ± 8.53	7.10 ** ± 9.35	−13.62 ** ± 16.05
Western Diet (38)	−1.90 ± 9.66	2.86 *± 14.24	−5.15 ** ± 7.57	−6.53 **± 9.03	7.78 ** ± 9.60	−12.82 **± 14.22
Mexican (10)	3.40 ± 11.83	13.04 * ± 17.65	−5.01 ** ± 3.45	−6.59 ± 11.83	11.94 ** ± 9.62	−14.35 ** ± 4.66
Italian (10)	−1.31 ± 3.10	4.46 ** ± 1.98	−2.95 ± 7.29	−8.11 ** ± 3.76	5.02 ** ± 2.75	−9.77 * ± 10.05
Mediterranean (9)	−6.80 ± 8.68	−3.11 ± 8.65	−10.10 ± 9.10	−8.94 ± 10.88	8.68 ** ± 6.48	−27.93 ** ± 16.92
American (9)	−3.53 ± 11.07	−4.27 ± 16.67	−2.81 ± 8.34	−2.28 ± 7.41	5.32 ± 15.27	0.58 ± 6.08
Eastern Diet (33)	−4.75 ** ± 10.10	1.03 ± 13.78	−9.99 ± 8.57	−12.15 ± 6.88	6.31 ** ± 9.14	−14.55 ** ± 18.11
Japanese (10)	−3.13 ** ± 2.20	1.92 * ± 2.86	−4.57 ** ± 1.61	−8.34 ** ± 3.76	9.44 ** ± 2.95	8.90 ** ± 3.65
Chinese (10)	−6.85 ** ± 2.31	1.83 * ± 2.87	−10.74 ** ± 1.57	−16.71 ** ± 2.09	3.65 ** ± 0.83	−20.62 ** ± 4.73
Korean (13)	−4.38 ± 16.07	−0.26 ± 22.16	−13.57 ** ± 12.38	−11.57 ** ± 9.09	5.94 * ± 14.20	−27.91 ** ± 12.88
Smoothie (22)	−9.13 ± 23.79	−7.70 ± 27.16	−9.86 * ± 21.40	−14.12 ** ± 23.53	−4.18 ± 25.22	−17.89 ** ± 27.84

Note: M: mean; SD: standard deviation; *n* ≤ 30 used Signed-Rank test * *p* < 0.05; ** *p* < 0.001.

**Table 3 nutrients-14-03168-t003:** Significant factors contributing to the differences between the internet-based application against Nutrition Data System for Research on total calories.

Parameters	Logistic Regression Original Model		Generalized Regression Elastic Net Model Validation	
	Estimate (95% CI)	*p* (*χ*^2^)	Estimate (95% CI)	*p* (*χ*^2^)
(Intercept)	3.1970 (1.8086, 4.5855)	<0.0001	3.1367 (1.7804, 4.4933)	<0.0001
<1000 caloric range	−1.4626 (−2.7979, −0.1274)	0.0318	−1.4136 (−2.7006, −0.1265)	0.0313
Fat % Difference	−3.3411 (−4.6729, −2.009)	<0.0001	−3.2879 (−4.5876, −1.9883)	<0.0001
Protein % Difference	−1.3791 (−2.4530, −0.3052)	0.0118	−1.3600 (−2.4073, −0.3127)	0.0109
MR	0.1852		0.1852	
AICc	30.29		30.29	
AUC	0.8920		0.8920	

Note: MR: misclassification rate; AICc: Akaike’s information criterion with corrections; AUC: Area under the curve; CI: confidence interval.

**Table 4 nutrients-14-03168-t004:** Significant factors contributing to the differences between internet-based application against Nutrition Data System for Research on folate.

Parameters	Logistic Regression Original Model		Generalized Regression Elastic Net Model Validation	
	Estimate (95% CI)	*p* (*χ*^2^)	Estimate (95% CI)	*p* (χ^2^)
(Intercept)	11.4831 (−139.76, 162.72)	0.8817	10.3903 (9.0613, 11.7193)	<0.0001
<1000 caloric range	1.4607 (0.4860, 2.4354)	0.0033	1.4604 (0.4854, 2.4354)	0.0033
Carbohydrate, % difference	−1.1903 (−2.9901, −0.9379	0.0001	−1.9187 (−2.8978, −0.9395)	0.0001
Italian diet	−11.2367 (−162.48, 140.00)	0.8842	−10.1439 (−11.3112, −8.9767)	<0.0001
MR	0.2963		0.2963	
AICc	38.52		38.52	
AUC	0.9046		0.9046	

Note: MR: misclassification rate; AICc: Akaike’s information criterion with corrections; AUC: Area under the curve; CI: confidence interval.

**Table 5 nutrients-14-03168-t005:** Significant factors contributing to the differences between internet-based application against Nutrition Data System for Research on cobalamin.

Parameters	Logistic Regression Original Model		Generalized Regression Elastic Net Model Validation	
	Estimate (95% CI)	*p* (*χ*^2^)	Estimate (95% CI)	*p* (*χ*^2^)
(Intercept)	−23.3204 (−314.45, 267.81)	0.8752	−5.0243 (−6.7635, 3.2852)	<0.0001
Protein, % difference	−3.2596 (−4.4292, −2.0901)	<0.0001	−2.6890 (−3.6172, 1.7607)	<0.0001
American diet	12.3545 (−207.66, 232.37)	0.9124	2.8874 (1.3699, 4.4048)	0.0002
Japanese diet	−23.3204 (−117.56, 203.75)	0.8752	3.8314 (2.7588, 4.9040)	<0.0001
MR	0.2593		0.2593	
AICc	37.73		37.42	
AUC	0.8083		0.8083	

Note: MR: misclassification rate; AICc: Akaike’s information criterion with corrections; AUC: Area under the curve; CI: confidence interval.

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
