# Peer review of "Validating Accuracy of an Internet-Based Application against USDA Computerized Nutrition Data System for Research on Essential Nutrients among Social-Ethnic Diets for the E-Health Era"

_nutrients, 2022, doi:10.3390/nu14153168_

Round 1

Reviewer 1 Report

Overall Comments: The authors compared the validity of mHealth diet data with NDSR. This topic is very interesting, and I appreciate the emphasis on diverse types of diets. However, the presentation of the manuscript requires more work. The main message is often overlooked due to how the manuscript is structured. Further, there is a lot of data presented using several types of analyses. Recommend structuring the paper so the audience can follow the results.

There were also errors surrounding nutrition concepts. For example, there are no carbohydrate-, protein-, fat- based micronutrients. NDSR is also a separate database than USDA. The methods need to be expounded upon to improve possible replicability, and the discussion requires a deeper interpretation (see comments).

ABSTRACT

·       Major Comments

o   Needs more description of methods in abstract (e.g., percentage of what)?

o   USDA and NDSR are two different approaches. NDSR uses USDA but are two different databases. Please revise.

o   Unclear what source factors mean.

o   Personalized nutrition also randomly appears and is not entirely pertinent to the study. The purpose of validating the mobile app is to improve tracking. There is a major leap from tracking to precision nutrition, especially as the study does not study how the data can be used for precision nutrition.

·       Major Comments

o   Unclear what protein and/or fat-related nutrients mean as that is not a common terminology in the nutrient field. Recommend removing those terms, as well as carbohydrate-related nutrients.

INTRODUCTION:

Major Comments:

·       Need more support for why we need to investigate other diets in the introduction.

·       Lines 44-47: The percentages reported (as well as the ICCs) need to be oriented. For example, “ranging from very strong for calories (intraclass correlation coefficient = 0.90 -1.00) …”

·       Require more explanation of what source factors truly mean and why certain factors were important to look at (thus, setting the stage for your methods).

METHODS:

Major Comments:

·       Lines 58-59: What is the difference between liquids and smoothies?

·       Section 2.1: Move the background on dietary patterns to the introduction. The methods should only discuss the methods.

·       The USDA is not NDSR.

·       Was a weekend date used for the 3-day diet records?

·       Line 165-166: Why were those caloric ranges used? Why did the authors choose those specific source factors?

·       Please provide more explanation about the food logs. Also, how adherent were the participants to tracking these apps? Did the recalls match the app days?

·       I’m still left confused for how these diets were categorized from the records? How were they matched to the app?

Minor Comments:

·       Line 72: Unclear why NIH was abbreviated and not the DRI?

·       Please remove mention about carbohydrate-related, meat-based. Vitamin A is found in liver.

RESULTS:

Major Comments:

·       What was the sample size?

·       The presentation of results is confusing, especially with the subgroup analysis. Further, it is easy for the reader to be lost with the combination of results from the Bland-Altman plots and Pearson’s. Please reorganize so it is easier for the reader. Also, please correct the name of the analysis).

·       Why were ICCs not used?

·       Recommend making it clear what analysis was performed with which analyses. As a reviewer, I would recommend identifying what the major thread of the manuscript should be. Right now, it reads very confusing as I feel the scope of the manuscript is way too large, especially with all the different types of analyses.

DISCUSSION:

Major Comments:

·       The discussion leads me with several questions. Why do the authors feel that the B-vitamins were overestimated? It seems that calcium from the Pearson’s and results from the Bland Altman somewhat contradict. Which results would you support?

·       Please provide further explanation for why you feel these sources presented as major contributors to differences in the measures.

TABLE 1:

·       Ok

TABLE 2:

·       Ok

TABLE 3:

·       For the logistic regression and others, please include the confidence intervals.

Reviewer 2 Report

Yang and colleagues have submitted a manuscript on an interesting and original topic. The authors present their results to validate an internet app for the purpose of estimating macro and micronutrients in various types of diets. I would like to highlight the large amount of data that was analyzed. The manuscript has areas that can be improved. Specific comments follow:

1.     Is there any reason why authors decided not to include iodine in the analysis?

2.     Page 2, Lines 54-55: Please improve sentence construction “he Bland-Altman analyses is commonly used methods to examine the validity of a 54 new measure [8,14,17-20].”

3.     Material and Methods: Typographical error, please amend heading “2.1. Social-Ethic Diets” 

4.     Material and Methods: Section 2.1 in one continuous piece of text, I suggest you divide into paragraphs, in general the information presented needs to be better organized to improve coherence. 

5.     Page 3, Line 98: Please amend “to patients of frailly aged,”.

6.     Material and Methods: Authors state “We analyzed 3-day dietary intakes of nutrients from social-ethic diets using both 120 NDSR and an app.”

·      Could you specify what criteria were used for the diet selection and name what social-ethnic diets were included?
